# People and Places: The Contextual Side of Politics in Demography and Geography

**Tadeusz Kugler [1],\* and J. Patrick Rhamey [2]**

[1] Department of International Relations and Political Science, Roger Williams University, One Old Ferry Road, Bristol, RI 02809, USA

[2] Department of International Studies and Political Science, Virginia Military Institute, 511 Scott Shipp Hall, Lexington, VA 24450, USA

\* Correspondence: tkugler@rwu.edu

**Abstract:** The disciplines of political demography and geography examine the interplay between social behaviors, spatial dimensions, politics, and policy. Investigations into demographic shifts, driven by evolving social norms or domestic and international political events, can influence numerous critical dependent variables in international relations, such as trade, development, and inter- and intra-state conflict. Similarly, geography and the interconnection of space with independent variables, such as power, wealth, and culture, yield similar insights. In this article, we employ a systemist approach from the Visual International Relations Project (VIRP) to provide a brief overview of the theoretical intersection between geography, demography, and international relations focusing on using VIRP to teach these subjects. To accomplish this, we have selected two representative pieces of literature from each field. For demography, we examine Hendrik Urdal's *A Clash of Generations? Youth Bulges and Political Violence*, and for geography, we review Alex Braithwaite's *The Geographic Spread of Militarized Disputes*. These seminal articles in their respective fields demonstrate the clear applicability of demography and geography to international politics scholarship.

**Keywords:** conflict management; demographic transition; geographic spread; Militarized Interstate Disputes (MIDs); youth surge

## 1. Introduction

Political demography and geography are among the most vibrant areas of scholarship in the disciplines of International Relations (IR) and Political Science (PS). The articles by Braithwaite (2006) and Urdal (2006) represent these two popular and growing research agendas. Urdal's work focuses on political demography, while Brathwaite's research is grounded in political geography. Both papers lay a foundation for future studies that examine the spatial influence of human existence and the age structure of populations on the likelihood of political violence. The geography of humanity is directly linked with its demography, as the physical makeup of the world creates the locations of habitation as well as the spatial realties of politics. The following are summaries of the two research fields and the articles presented through a systemist lens. Systemism, as outlined in the introduction to this special issue and revisited later on, emphasizes an approach that uses mutually intelligible diagrams that depict respective arguments and thereby improve communication in the academic world (Gansen and James 2021, 2023).

Political demography investigates the intricate relationship between demographic factors and political outcomes. By incorporating concepts from demography, political science, sociology, and economics, this field analyzes the influence of population dynamics—such as age structure, fertility rates, migration, and urbanization—on various political phenomena. Key areas of inquiry include the association between age structure and political violence, with particular attention to the role of youth bulges in contributing to political instability, civil conflict, and terrorism, as well as the mitigating or exacerbating

effects of factors such as education, employment opportunities, and political institutions. How and when violence starts or is expected to start, its onset, the types of action, whether they be civil disruption, such as riots, or actions with higher casualties, such as terrorism, is an important component of study. Additionally, political demographers examine the impact of migration on politics, analyzing how immigration and emigration influence political participation, social integration, public opinion, and policy-making in both the sending and receiving countries. Lastly, the research agenda encompasses the study of fertility rates, family planning policies, and population growth, investigating their effects on political, social, and economic outcomes, including gender equality, resource allocation, and policy priorities.

Political geography, an increasingly prominent research agenda within IR and PS, explores the complex interplay between geographical factors and political processes, outcomes, and institutions. This interdisciplinary field examines how spatial dimensions, such as territorial boundaries, resource distribution, and physical landscapes, influence political behavior, power dynamics, and state interactions. Central areas of study in political geography include the impact of territorial disputes on conflict and cooperation, the role of regionalism and local identities in shaping political preferences and outcomes, and the influence of resource distribution and environmental factors on national and international politics. Additionally, political geographers analyze how global processes, such as globalization and climate change, affect political decisions and interactions at various levels, from local to international. By adopting a spatial perspective, political geography offers valuable insights into understanding political phenomena and contributes to the development of more comprehensive theories and policy solutions.

Because understanding the impact of space and population on political phenomena is conceptually and technically complex, the systemist visual method of Gansen and James (2021, 2023) provides a means to parse out the causal mechanisms of people and place in relation to political outcomes, including inter- and intra-state conflict. Systemist notation is followed in each figure and a full explanation of it appears in Gansen and James (2023). Text in each figure is typed in UPPER- or lower-case characters. UPPER case characters are used for MACRO variables, while lower case characters are used for micro-level variables. Each diagram also comes in double frames—the outer one refers to the environment, the inner one to the system. The featured works focus on the discipline of IR as the system, with the World Beyond as its environment. The macro and micro levels of IR, respectively, correspond to the discipline as a whole and individual scholars within it.

Implementation of systemist graphics in a publication facilitates the relating of its claims to research across not only the geography and demography subfields but also broader IR and PS. For example, findings on the diffusion of conflict and the theoretical processes underlying these results can, and should, be integrated into more traditional dyadic, quantitative conflict analyses to offer more accurately specified models of conflict behavior. The systemist method grants clarity about the causal process, uncovering the variables that matter and in what contexts. Is resource presence a necessary control? What types of territoriality or contiguity offer insights into conflict opportunity and willingness? How should age distributions be included in our theories of international conflict? Graphic depictions such as those included herein allow for easy translation of material from one subfield to a relevant other, both broadening and deepening our causal understanding of political outcomes.

## 2. Population and Politics

### 2.1. The 'Youth Surge'

Rapid population growth is typical in the developing world in the post-World War II era (Kugler 2023; Sciubba 2011, 2022). Scholars from a wide range of fields study the importance of population change and movement: influence on great power status, the importance of aging, effects on international policy, or how new age cohorts affect the stability of countries (Furlong and Eck 2018). This last query is the empirical question of

Urdal (2006). One of the first projects that linked the new world of nations with high youth populations was called a 'youth surge', with a focus on conflict within those nations. Does a large relative amount of youth increase the chances of violence or attempted political change within nations? As this phenomenon is global, with nearly every developing nation on earth having a youth surge, the importance of the project is evident, as is the difficulty of moving from demographic statistics to a causal theory underlying observed events. Causality is of prime importance but has remained an issue in studying population's relationship to politics.

Population as a core component of power, particularly relevant to the study of global or regional rivals, underlies research agendas under the umbrella of 'power transitions' starting with Organski and Kugler (1980), and measurement of power in Waltz (1979). Population was then integrated with concepts of government capability to understand the importance of regional hierarchies in Organski et al. (1984) and later (Kugler 2023) considered how demography generally could be altered by unprecedented population size and growth.

The scale of that growth following World War II was so large that it became an interesting and critical question about how humanity can operate. Goldstone et al. (2012) focused on security concerns, starting with growth and then leading to aging. Weiner and Teitelbaum (2001) attempted to create a more specific theoretical foundation for the demographic literature, with Cincotta et al. (2003) concerned with the possibilities of political change in post-communist states. Research agendas considering whether this new population would facilitate economic growth grew from the phrase 'the demographic dividend'. They focused on the new modern world of massive expansions of a nation's labor force as generations enjoyed rapid declines in infant mortality and remarkable population growth. This youth surge was expected to propel a country to new heights of economic development. Importantly, this agenda does not deny the possibility of the increased potential for conflict, but only that youth itself can cause various disruptions, from economic growth to violence. Youth, at its core, represents potential that can be utilized in either direction, or even concurrently.

Looming over these considerations is yet another aspect of modern demography: the aging of societies. This process of moving from a society in which the largest proportion of the population is young to one rapidly aging is called the 'demographic transition' or 'silver tsunami' (Stowell 2021). Part of the process is seen in the 20th century with its rapid health innovations causing substantial declines in all forms of mortality leading to the largest population growth ever recorded. The term 'youth surge' comes from this surge of population that unexpectedly survived to adulthood but is disproportionately large when compared to the age structure of society historically. The consequences are unprecedented, with the youngest generation being the smallest now that humanity lives well beyond its expected working years. How to feed, pay, and provide for individuals who expect to enjoy nearly twenty years of retirement is a unique societal conundrum. This is again seen in the demographical statistics, but the 'why?' of how this process can take a standard form remains unanswered. The process looks much the same across countries, from those with high economic success to the mediocre, those controlled by traditional concepts of family, to those in a post-modern world.

### 2.2. Youth and Violence

*A Clash of Generations? Youth Bulges and Political Violence* (Urdal 2006) is a seminal contribution to studying the relationship between demographic factors and political violence. In the article, Urdal argues that "youth bulges", or an excess of young people in a population, can increase the risk of political violence. The systemist diagram appears as Figure 1.

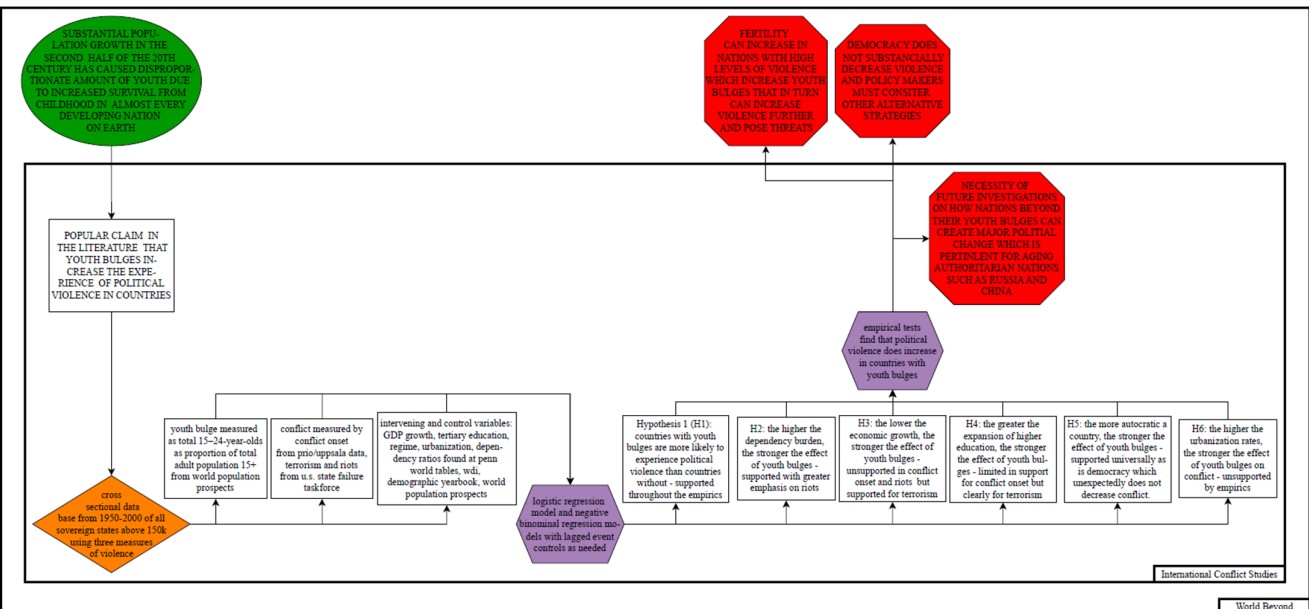

**Figure 1.** *A Clash of Civilizations? Youth Bulges and Political Violence* (Henrik Urdal 2006). Diagrammed by: Tadeusz Kugler, Sarah Gansen and Patrick James.

Figure 1 illustrates that empirical analysis lies at the heart of Urdal's (2006) investigation of the youth bulge phenomenon. Analysis starts with the initial variable illustrated within the green oval. Here, the phenomena of decline in mortality leading to youth surges is linked to the generic variable of and foundation of the paper—that youth surges could create violence. To address this research interest, Urdal creates an additional linkage and constructs a cross-national dataset spanning 1950 to 2000. This dataset is further linked to measures of violence in three different ways, incorporating alternative explanations into the empirical analysis, which are then articulated in the hypothesis statements.

One critical component is the dependent variable, where Urdal diverges from much of the prior literature on the subject. Focusing on 15–24-year-olds as a proportion of the total population aged 15 and older, this study invites debate on whether the youth bulge should be measured as Urdal suggests or through dependency ratios or gender segmentation. Regarding conflict, Urdal employs the widely recognized PRIO/Uppsala conflict database (Gleditsch et al. 2002). To account for alternative sources of violence, classic control variables from conflict and international political economy literature are utilized as specified in the graphic. This comprehensive approach ensures that Urdal's analysis accounts for the multiple dimensions that may contribute to the youth bulge phenomenon and its potential link to violence. Each of these subsections of the data are generic variables within the outlined boxes in Figure 1.

The nodal linkage between data and hypothesis is the use of negative binominal and logistic regressions to connect evaluation of six hypotheses through eleven distinct models. Let us first discuss the outcomes of the hypothesis that as a group form multiple pathways. Note that the purple hexagon suggests that youth surges do increase violence. Then we can move to the end point of the relationships within the red octagon, which are future research agendas and policy recommendations.

Beginning with Hypothesis 1:

**Hypothesis 1.** *Countries that experience youth bulges are more likely to experience political violence than countries without.*

Models 1 and 2 (Urdal 2006, pp. 617–18) show consistent support for H1, in which countries undergoing youth bulges exhibit more significant degrees of violence. This outcome has a greater degree of robustness, as it retains its consistency even with various

control variables used to parcel out the significance, as seen in Model 3 using differing measurements of GDP and in Models 4, 5, and 6, all of which help illustrate the importance of evaluating a choice as to what measurement of a youth bulge is appropriate. Choices related to gender are highlighted by previous scholars concerned with the disproportionate male population, whereas Urdal (2006) uses the total youth population. Interestingly, these previous studies showed no support for youth bulges being linked to violence when measured as a percentage of males to total population by age cohort. This finding can be a helpful foundation for future research.

An additional finding that prompts future research is that economic growth or its absence does not seemingly increase the potential for conflict, or at least is of such limited substantive influence as to be unimportant. This is both a surprising and exciting finding, given the standard assumptions of economic ineffectiveness leading to widespread unrest seen in Models 7 and 8. At least in these models, the foundational demographic of the youth bulge itself matters more than the economic environment. Model 9 considers the type of government, in this case showing a slight increase in violence under autocratic regimes with high levels of tertiary education growth, though substantively limited in its positive influences on increased violence in countries with a youth bulge.

**Hypothesis 2.** *The higher the dependency burden which is the expected amount that society is allocating resources to the youth, the stronger the effect of youth bulges on political violence.*

H2 is supported by the empirical evidence mentioned earlier. The more critical component is which type of measurement to use for political violence: terrorism or riots. The dependency ratio is strongly associated, first with lower violence, measured by riots, then when linked to other factors, such as how higher education influences terrorism.

**Hypothesis 3.** *The lower the economic growth, the stronger the effect of youth bulges on political violence.*

Supported by the models, but unlike in H2, this is more prevalent in the terrorism-measured violence models as opposed to the riot model. Interestingly, this does not increase the chances of armed conflict, only the increase in terrorism as an outcome. Again, this may be a valuable foundation for future research.

**Hypothesis 4.** *The greater the expansion of higher education, the stronger the effect of youth bulges on political violence.*

There is limited support for H4 regarding the start of political violence, but a clear positive linkage with terrorism as a subcomponent.

**Hypothesis 5.** *The more autocratic a country, the stronger the effect of youth bulges on political violence.*

H5 is supported, as violence is positive for authoritarian societies, as would be expected, but also unexpectedly positive for democracies. This finding is of particular concern given that foreign policy recommendations on alleviating domestic disturbances via increased voting would lead to a surprising result of violence being abated only limitedly. The underlying demography's influence outweighs the political system's importance.

**Hypothesis 6.** *The higher the urbanization rates, the stronger the effect of youth bulges on political violence.*

No apparent effect on violence is revealed by any metric, an interesting outcome that demands further study. The scale of new urban environments with the creation of

mega-cities might be a fruitful avenue for future exploration that involves changing the metric, but not the hypothesis.

Urdal suggests that youth bulges create a range of social, economic, and political pressures that can lead to violent conflict. Specifically, he argues that young people in countries with youth bulges may face limited economic opportunities, limited access to education and healthcare, and increased political marginalization. These factors, in turn, can create frustration, hopelessness, and anger that can contribute to political violence.

Urdal's article is notable for its empirical rigor, as he uses statistical analysis to demonstrate the relationship between youth bulges and political violence. He also identifies several important policy implications of his research, such as investing in education, job training for young people, and the need to promote political inclusion and participation. Overall, Urdal's article is an important contribution to conflict studies, providing a nuanced and empirically grounded understanding of the complex relationship between demographic factors and political violence. His work has inspired further research on this topic and has helped inform policy discussions on mitigating the risk of violent conflict in countries with youth bulges.

The 'how?' and 'why?' of these events need to be examined in future research, such as that suggested in the red octagons from Figure 1. First, questions of how to change the politics of nations outside the youth bulge are of tremendous interest. China, Russia, and nearly all of Latin America are nations and regions that are no longer relatively young. How might authoritarian nations be expected to change, given their increasingly aging societies? Russia is quickly becoming an aged nation; if youth in needed for political change, does this trap the preceding nations in dictatorships (Stowell 2021)?

Following on from the preceding queries, the most challenging component of social science is understanding causality. What would be the framework by which youth is more likely to create political violence? Is it the expectation borne out, from the ancient Greeks onward, that youth is more willing to accept risk for change? This classic idea of risk-acceptant behavior is seen in work on criminality, public health, and even automobile accidents.

Countries with youth bulges, from Iran to Nigeria, Yemen to Pakistan, are reasonably common in the developing world. The question of causality again rears its head. Is the youth bulge, created not by happenstance, but by the combination of increased survival rates of infants via the introduction of cheap medicines from the global community, linked to traditional family concepts that required large amounts of children to sustain the household?

The first and most common response to that multifaceted question is that limited economic growth increases fertility rates, leading to a painful causality loop. Second, the danger of this would be the experience of a 'poverty trap', in which conflict decreases economic performance and increases fertility, which in turn increases conflict. This process creates a poverty trap in which each generation has less than the one before, with the only consistent change being the increase in total population. Notice, therefore, that Urdal (2006) shows that economic prosperity is not the key to understanding political violence. Instead, youth are more of a factor that complicates the scenario and should lead in the future to exciting research on multigenerational causality regarding political violence, and how to break that chain. Third, and last, are the limited measured effects of democracy. A regime that continues to be the standard preferred system of governance is not necessarily a panacea to pull youth from violence into economic prosperity. Limitations on how the international community can revise domestic policies to unlock demographic potential are significant.

Demography continues to be an understudied component of politics, but works like that of Urdal (2006) help to enlighten this crucial correlate of political outcomes. The empirics of the article are an essential contribution to the discussions on how and where political violence would be expected to occur. The influences of education and regime on

those expectations, particularly in terms of seemingly skipping riots and moving directly towards terrorism, and the limited abatement effects of democracy are of particular concern.

### 3. Geography and Politics

*3.1. Geographic Analysis*

Like demography, political geography confronts questions of context: where does politics occur? Just as analysis of demographic trends may examine migratory flows or internal birthrates, geographic analysis may be externally or internally defined. Internal analyses examine how terrain and distance affect civil war (Linke and Raleigh 2016). External examination of geography may focus on how the distribution of power across space by the most powerful states affects interstate conflict behaviors (Lemke 2002; Rhamey et al. 2015), or how domestic and international dynamics are linked across geographic space in order to explain political outcomes (e.g., Buhaug and Gleditsch 2008; Solingen 2012; Ward and Gleditsch 2002).

While an emphasis on geography in the study of politics is not new (see, for example, Mackinder (2004), the rising interest in geographic space as a relevant contextual and potentially causal consideration to political outcomes follows on the heels of comprehensive dyadic analysis of state behavior (Miller et al. 2010). Before political geography's growth, dyadic research often ignored location altogether or treated it as a nuisance to be dealt with in case selection through the identification of politically relevant dyads, though not without potential pitfalls (Lemke and Reed 2001). Serious substantive emphasis on space has risen to prominence over the last two decades, prompted by a series of studies such as Braithwaite's seminal contribution on conflict dynamics, which is focused upon here. This study then prompted a growing body of research (e.g., Gibler and Braithwaite 2013), including the development of geo-focused MID data that treats geographic space as fundamental to our observations of conflict between states (Braithwaite 2010).

*3.2. The Spread of Conflict*

Rather than simply examining conflict's presence (see Braithwaite 2005), the article *The Geographic Spread of Militarized Disputes* (Braithwaite 2006) focuses on the conditions affecting conflicts' spread. This contagion effect is a byproduct of political conditions contingent upon geographic considerations, resulting in either the ease of conflicts' spread or its significant limitation. Importantly, the relevance of these geographic limitations only makes sense in the context of the political lens. For example, mountains, resources, or borders matter so far as political decision-makers perceive them as salient. The dependent variable builds on Buhaug and Gates (2002); the geographic spread of MIDs is operationalized as a circular area measured in $km^2$—thereby creating a geographic construction of the data.

Graphic depiction of Braithwaite's analysis in Figure 2 identifies the relevant "world beyond" as the realm of politics generally. The micro level emphasizes the need for scholars of IR to account for geographic context in examining political phenomena. In that world beyond, policy analysts and policymakers should account for geography in the practice of politics, perhaps improving the efficiency of conflict management and military deployments by states. Further, geography's relevance is inherently tied to people's experience of how the conflict unfolds, allowing for a better understanding of a conflict's impact beyond simply onset or fatalities.

To examine the geographic spread of conflict's impact, Braithwaite operationalizes the dependent variable as the distribution of a dispute's engagement in squared kilometers in a circle around the mean location of the conflict. The examination of the dependent variable begins with an examination of space in political relevance and terrain, mixed with the geographic salience of the belligerents. On the political nature of space, is the dispute over territorial issues, including the presence of a shared border? On politically relevant characteristics of the terrain, does the space include key natural resources, and is it easily traversed? Finally, are the states themselves large (in geographic size)?

Findings that answer these broad questions are important to the political geography literature and to some commonsense expectations about geography's relevance to conflict contagion, e.g., a state must have something to fight over, such as resources. However, the preliminary results are limited in scope. Territorial issues do matter, consistent with established expectations about the origins of most interstate conflict (Diehl 1992). Furthermore, resource presence also increases the spread of strife. However, interestingly, vital borders *decrease* the contagion.

Therein lies an interesting tension in the research's initial findings: territorial issues, wherein at least one party to a conflict expresses a reason for the war as a territorial matter, increase the spread; however, if more specifically determined, and the geographic space contains a vital border that is of importance and salient to one or both parties, the potential for spread is reduced, hinting at the potential for some risk aversion by states dealing with salient border-related geographies. In this initial model, the politically relevant aspects of geography (resources, territorial issues, borders) do much of the heavy lifting in understanding conflict's geographic spread. The dimensions with what could be described as a more agnostic character, such as terrain or state size, offer less explanatory power.

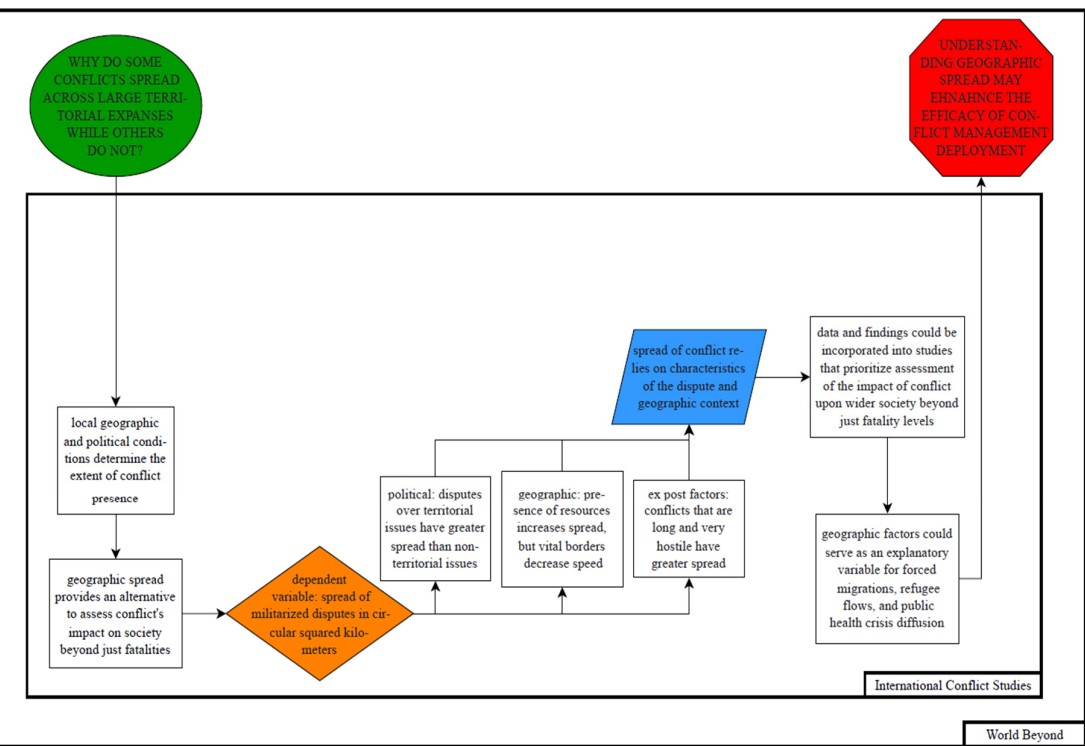

**Figure 2.** The Geographic Spread of militarized Disputes (Braithwaite 2006). Diagrammed by: J. Patrick Rhamey, Jr., Sarah Gansen and Patrick James.

To better contextualize the nature of these geographic relationships, a second empirical test reveals the relevance of ex-post information to better understand diffusion of conflict. While including ex-post information may be limiting to policymakers in the "world beyond" who do not have this ex-post information at their fingertips, it does allow for further refinement in understanding the spread's potential sources. If a conflict is longer and more deadly, it spreads. While we may not be able to anticipate whether conflicts will endure or increase in casualties ahead of time, this finding does provide incentives to limit conflict duration: the longer a conflict endures, the more geography it covers, and therefore, the more people it impacts across that contextual geographic space.

Lastly, Braithwaite connects the significance of this geographic diffusion to the interplay between people and place, emphasizing the link between geography and demography. The potential for conflict diffusion extends beyond those in the geographically contested

area. The systemist diagram underscores the intrinsic connection between geography and demography: as conflicts widen to encompass more people within the conflict zone, the implications for those beyond the zone also increase, shedding light on potential migratory, economic, and resource-related concerns.

*3.3. Linking the Diagrams Together: Demography Meets Geography*

Use of systemist notation to depict the two articles helps better identify the underlying causal processes when examining political phenomena in the demographic and geographic context. Both articles are representative of broader quantitative investigations in their respective subfields, with many moving parts and multiple competing hypotheses. The clear, sequential examination of causal processes and findings provided by systemist notation allows for a better understanding of the articles' arguments and the verification or refutation of competing hypotheses. Tackling complex contextual questions about space and population can be made more simple, understandable, and comparable to other related research when using the systemist method.

Comparing the two notations offers some interesting insights into how the subfields of political demography and geography are closely related. Given political geography's interest in the political experiences of people in space, youth populations, as described by Urdal (2006), have consequences for the broader questions of conflict examined by Braithwaite (2006). To what extent do demographic shifts or youth dividends create the instability that may lead to significant clashes over territory, not just within regimes but beyond them? While Braithwaite finds that conflicts spread, as Buhaug and Gleditsch (2008) note elsewhere, conflicts may also lead to other conflicts. The degree to which conflict contagion occurs is likely related in part to internal political demography.

Of course, this is just one example of a newly fruitful research agenda. Not only do demography and geography relate to one another, but they also have important consequences for other subfields in IR and PS, including institutions, development, political behavior, and foreign policy analysis. A first example of future research linking the two is the concept of prime locations, these being the geography of where the population is most likely to want to live, most often near the tropics, but with technological advancements moving towards colder climates. The role of climate change on the malleability of primary locations is significant, as increased heat could force population from the center of humanity, which is near the equator, towards locations that have not seen such levels of population density, from the steppes of Asia to the plains of Canada. Within conflict studies, questions of why insurgencies have had success can be tied to mountains or covered terrain, such as jungles. Mountains are clearly not dynamic, but their very existence has seen a heavy focus as they are an advantage in wars against more technological adversaries. The rise of new megacities is itself both demographic and geographic and could be the extension, as humanity builds structures housing over fifty million people, of the cover and terrain seen in mountains or jungles. These new studies combining expectations of migrations from warmer environments to the geographic stability of colder climates are exciting prospects of future study. What are the possibilities of creating new locations of human centralization vs. historical happenstance is also an important future study. Systemist analysis highlighted by the two graphics represented in this paper allows for drawing lines, not only within a single work's causal process but also, through clarity of communication, facilitates the drawing of substantive connections across both individual articles and broader subfields.

## 4. Conclusions

Urdal (2006) and Braithwaite (2006) represent significant contributions to international conflict studies, as they are foundational papers integrating measurements of demography and geography into the empirical analysis of violence. Findings reveal that a nation's cohort age characteristics indeed influence the propensity for violence, a crucial insight when considering the fragility of much of the developing world. Furthermore, the nature of territory is essential in understanding the intensity of wars and how this interconnects

with the demographic composition within regions. States with young populations, limited border controls, and festering territorial disputes are ones with very high levels of conflict likelihood. Ethnic, national, and migrational interests forge connections between these two papers and lay the groundwork for future research that combines demography and geography under a unified, sustainable, and systematic research agenda.

**Author Contributions:** Writing—original draft preparation, T.K. and J.P.R. Writing—review and editing, T.K. and J.P.R. All authors have read and agreed to the published version of the manuscript and are equal contributors.

**Funding:** This research received no external funding.

**Institutional Review Board Statement:** Not applicable for this paper or the papers reviewed.

**Informed Consent Statement:** Not applicable.

**Data Availability Statement:** Data is contained within the articles of Urdal 2006 and Braithwaite 2006.

**Conflicts of Interest:** The authors declare no conflict of interest.

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
