# Peer review of "People and Places: The Contextual Side of Politics in Demography and Geography"

_socsci, doi:10.3390/socsci12080456_

Round 1

Reviewer 1 Report

This article on ‘people and places’ is well written and achieves its aim: I was not familiar with the subject but feel I could grasp it clearly from the presentation. This of course is the core goal of the article using the systemist approach. I have only a few minor suggestions:

1.      Lines 11-22:

a.      Topic DV, but then discussion of IV (confusing).

b.     22: ‘neg. bin. & log’ regression: maybe explain distinction of onsets, terrorism, and riots

2.      53-55: are interactions being described?

a.      What is ‘dependency burden’? Interacted with youth bulges?

b.     Finding in wrong direction? How linked?

3.      Mix-up 68-69

4.      75: “urbanization rates”: is this a static (percent) or dynamic (change) measure?

5.      157: Is this a section title?

6.      Misspelling in stop sign in Figure 2

Author Response

Thank you for your review it has been very useful for our editing. Let me get to some of your points. 

  1.      Lines 11-2.

We have edited and expanded on each of your suggestions. We do have a little bit of a space limitation but hopefully when the paper is taken as a whole why demography is so linked to geography will be illustrated by these two example papers. 

  1. 53-55: are interactions being described?

Dependency burden is the amount of research society is allocating towards a particular age cohort. It can be measured by the percent of the cohort in ratio to the working-age population and is a commonly used technique when aging societies are evaluated. In this case, Urdal is trying to pull the importance of the total amount of population, and youth surge, in comparison with it in ratio to the working-age population. Is there a possibility of causal pressure from the need for additional resources, or its demand, or is the age itself of the population the important characteristic?

"H2 is supported by empirical evidence mentioned earlier. The more critical component is which type of measurement used for political violence: terrorism or riots. The dependency ratio is strongly associated first with lower violence measured by riots, then when linked to others such as higher education, influences terrorism."

"Hypothesis 3: The lower the economic growth, the stronger the effect of youth bulges on political violence

Supported by the models but unlike in H2, this is more prevalent in the terrorism-measured violence models as opposed to riots. Interestingly this does not increase the chances of armed conflict, only the increase in terrorism as the outcome of it. Again, this may be a valuable foundation for future research."

The outcome of his research suggests additional complexity between the two different concepts of youth. Firstly why they would be interested in violence, be it from limited economic growth or additional educational opportunities, and secondary does some combination of all four more likely lead to demonstrations, riots, violence, or terrorism? It is interesting finding the differences in direction and a reason for future study. 

  1. 75: “urbanization rates”: is this a static (percent) or dynamic (change) measure?

It is the percent of the total population which does change yearly in most countries it increases. 

  1. 157: Is this a section title?

Yes

  1. Misspelling in stop sign in Figure 2

Thank you for catching that!

Reviewer 2 Report

Having reviewed the criteria for the special issue, I think this article fits the criteria and has some merit. However, I believe there are a number of opportunities to improve the manuscript. Currently, the strength of the paper is in the summary of the two motivating articles. They are accurately and thoroughly discussed. The current manuscript does not do enough, though, to link the two together, or to emphasize the relationship between the two. Section 3.2 (the section that would move beyond a comprehensive literature review and actually make a contribution to future work) needs to be much more developed before I would recommend acceptance. There is currently one paragraph which clearly focuses on a discussion linking political geography and political demography. I recommend the author(s) be provided an opportunity to enhance that section and resubmit, as the foundation for a good contribution to the special issue is clearly there.

Author Response

We appreciate your review! We have added additional expectations of future research something we did not have in the original paper to that level primarily as we started to hit the word limit for the special issue and wanted to focus on how to read and use the graphics of the two papers in a teaching context. 

Reviewer 3 Report

This is basically a literature review article that focuses on the work of two scholars [Alex Braithwaite and Hendrik Urdal] on the aspects of geography and demography as factors contributing in the eruption and diffusion of conflict. The article analytically covers the two major works published by the aforementioned authors arguing for the need to link and thus jointly study the two aspects [demography and geography] in our quest of understanding the onset and geospatial spread of conflict. I find myself in agreement with the suggestion for an holistic overview of conflict with emphasis on the geographic and demographic factors conducive to the emergence and spread of violent conflict, but, as mentioned above, this is just a literature review article with a narrow and selective focus on the two aforementioned works, and it does not bring about any new contribution, analysis or insight of any meaningful type; be it in the form of producing fresh evidence through data analysis, criticism or evaluation of secondary data, or through empirical observations [e.g. fieldwork]. Rather, the article provides a mere synopsis of the two major works pointing to the merging of these two aspects. Given the complete lack of original research of any kind, or the development of theoretical discussion, it cannot be considered for publication as it currently stands. I would suggest using the two major works as starting points for developing an argument that would lead towards the inception of a theoretical model, a mechanism or approach that could be tested and used in further analysis. It could be thus reconsidered, as it contains a well-presented and valid theoretical overview, but only after it develops a sense of contribution and originality. 

Author Response

We appreciate your review and know the time it takes from busy summer schedules. This paper was designed to be part of a special issue which is investigating the use of illustrations in a teaching context as part of the Visual International Relations Project (VIRP) and the paper is not designed to be independent research. 

Round 2

Reviewer 2 Report

I thank the authors for their attention to my comments. 

Author Response

Thank you again for your time and effort on our paper. We hope you have an enjoyable summer!

Reviewer 3 Report

The article  may be published as a literature review piece of previous works.  It does not contain any original research or contribution, a sit is intended teaching and educational purposes, and this should be stated clearly in the article's abstract and introduction.    

Author Response

"The article  may be published as a literature review piece of previous works.  It does not contain any original research or contribution, a sit is intended teaching and educational purposes, and this should be stated clearly in the article's abstract and introduction."    

Thank you again for your efforts and we will highlight the teaching aspect in the abstract. We hope you have an enjoyable summer.